# General negative pressure annealing approach for creating ultra-high-loading single atom catalyst libraries

Yi Wang [1,8], Chongao Li[1,8], Xiao Han[2,8], Jintao Bai[1], Xuejing Wang[3], Lirong Zheng[4], Chunxia Hong[5], Zhijun Li [6], Jinbo Bai[7], Kunyue Leng [1]✉, Yue Lin [2]✉ & Yunteng Qu [1]✉

Catalyst systems populated by high-density single atoms are crucial for improving catalytic activity and selectivity, which can potentially maximize the industrial prospects of heterogeneous single-atom catalysts (SACs). However, achieving high-loading SACs with metal contents above 10 wt% remains challenging. Here we describe a general negative pressure annealing strategy to fabricate ultrahigh-loading SACs with metal contents up to 27.3–44.8 wt% for 13 different metals on a typical carbon nitride matrix. Furthermore, our approach enables the synthesis of high-entropy single-atom catalysts (HESACs) that exhibit the coexistence of multiple metal single atoms with high metal contents. In-situ aberration-corrected HAADF-STEM (AC-STEM) combined with ex-situ X-ray absorption fine structure (XAFS) demonstrate that the negative pressure annealing treatment accelerates the removal of anionic ligand in metal precursors and boosts the bonding of metal species with N defective sites, enabling the formation of dense N-coordinated metal sites. Increasing metal loading on a platinum (Pt) SAC to 41.8 wt% significantly enhances the activity of propane oxidation towards liquid products, including acetone, methanol, and acetic acid et al. This work presents a straightforward and universal approach for achieving many low-cost and high-density SACs for efficient catalytic transformations.

The development of advanced catalysts must meet the requirements of future sustainable chemistry, but their commercial potential is contingent on high reaction efficiency and maximal atom economy[1]. Single atom catalysts (SACs), integrating atomically dispersed metal centers with tunable coordination structures over appropriate supports, exhibit remarkable activity and unique selectivity in electrocatalysis, photocatalysis, and thermal-catalysis[2–7]. Moreover, the maximal atom utilization efficiency of this class of catalysts greatly improves the atom economy, especially for noble-metal-based catalysts. Therefore, it is beneficial for sustainable chemistry[8–12]. Given

[1]International Collaborative Center on Photoelectric Technology and Nano Functional Materials, Institute of Photonics and Photon-Technology, Northwest University, Xi'an, Shaanxi 710069, China. [2]Department of Chemistry, Department of Applied Chemistry, Hefei National Research Center for Physical Sciences at the Microscale, University of Science and Technology of China, Hefei, Anhui 230026, China. [3]Interdisciplinary Research Center of Biology & Catalysis, School of Life Sciences, Northwestern Polytechnical University, Xi'an 710000, China. [4]Institute of High Energy Physics, Beijing 100039, China. [5]Shanghai Advanced Research Institute, Chinese Academy of Science, Shanghai 201210, China. [6]National Key Laboratory of Continental Shale Oil, College of Chemistry and Chemical Engineering, Northeast Petroleum University, Daqing 163318, China. [7]Université Paris-Saclay, CentraleSupélec, ENS Paris-Saclay, CNRS, LMPS-Laboratoire de Mécanique Paris-Saclay, 8-10 rue Joliot-Curie, Gif-sur-Yvette 91190, France. [8]These authors contributed equally: Yi Wang, Chongao Li, Xiao Han. ✉e-mail: lengky@nwu.edu.cn; linyue@ustc.edu.cn; yuntengqu@nwu.edu.cn

these merits of SACs, tremendous efforts have been devoted to developing a variety of synthesis methods for many technical applications[13–16]. Nevertheless, considering their high surface energy, the SACs are generally constructed with low metal loadings to circumvent the aggregation of metal atoms into metal clusters or nanoparticles[17]. This results in a low metal areal density. Taking the Pt SACs as a typical example, they display impressive activity and selectivity in the thermal-driven activation of light alkane[18–20]. A literature analysis shows that the metal contents of most Pt SACs are 2 wt% or below (Fig. 1a and Supplementary Table 1)[21–24], and the Pt areal density of most catalysts are hard to surpass 1.5 atoms/nm² (Supplementary Table 1). In this case, the SACs with insufficient areal density of active sites not only limit their overall catalytic performance but also decrease the productivity per unit volume or mass of catalysts. Therefore, the development of a universal synthesis strategy for

accessing SACs with high metal loading and sufficient areal density is significant in this field, but challenging[25–28].

Recently, several strategies have been reported to construct ultra-high-loading (UHL, higher than 10 wt%) SACs on different supports[29]. Wang et al adopted crosslinking carbon quantum dots as supports to provide abundant anchoring sites to favor the formation of high densities of single metal atoms[30]. Lu reported a two-step annealing strategy to obtain high-loading SACs on distinct carbon and metal oxide supports[31]. This strategy effectively controls the bonding of metal precursors with the carrier and prevents thermal-induced aggregation of metal into nanoparticles. Zou developed a laser planting method to simultaneously create defects and anchor metal atoms, eventually achieving high-loading single metal atoms on carbon, TiO₂, and Cu NPs[32]. In contrast with these pioneering works, a facile and routine available synthetic strategy without the using of expensive

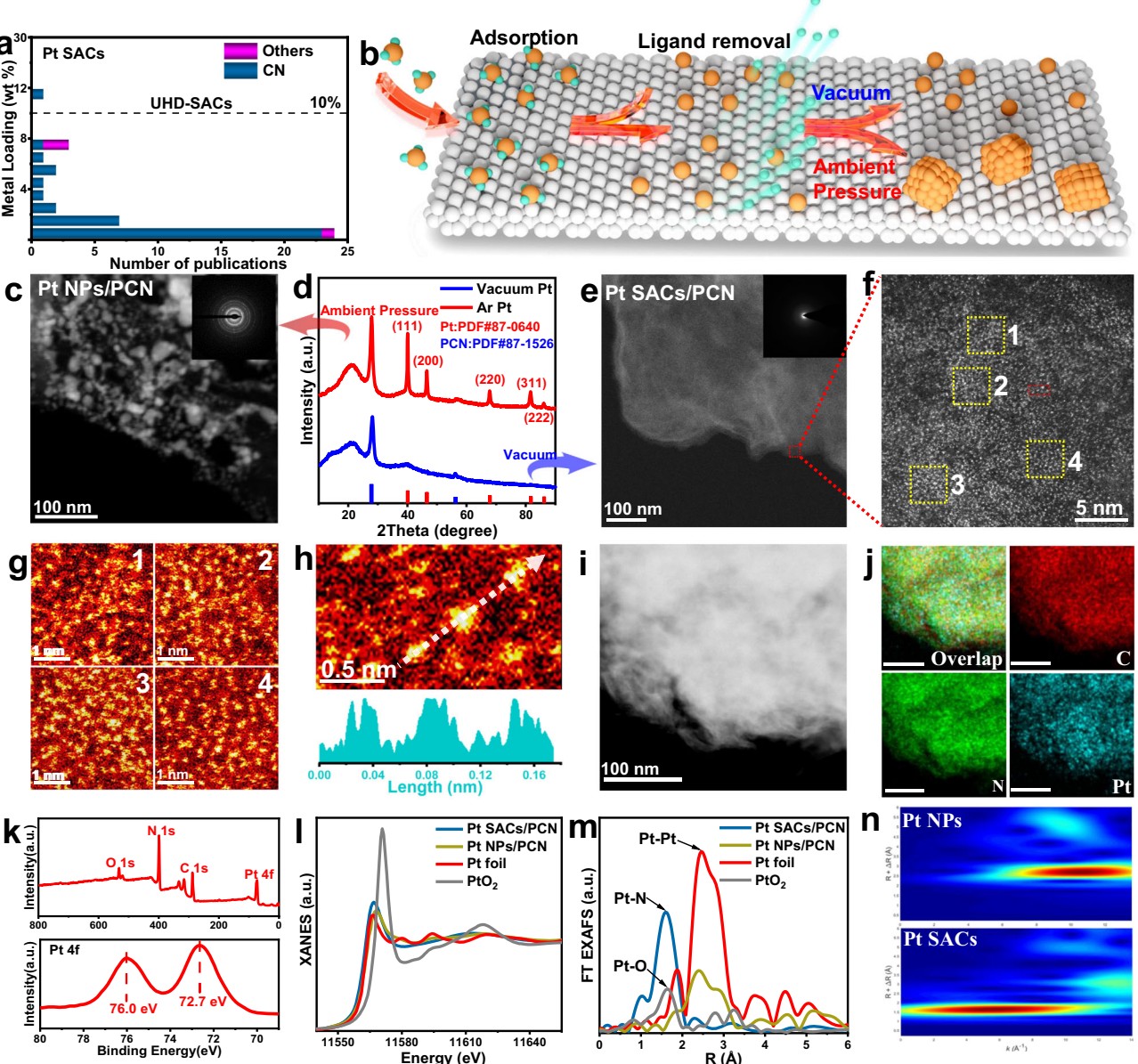

**Fig. 1 | Structure investigation of the Pt UHL-SACs (Pt SACs/PCN). a** Metal contents of the most reported Pt SACs. **b** Schematic illustration for preparing UHL-SACs. **c** TEM image of 40.9 wt% Pt NP/PCN. **d** XRD patterns of 40.9 wt% Pt NP/PCN and 41.8 wt% Pt SACs/PCN. **e–n** Structure characterization of 41.8 wt% Pt SACs/PCN: (**e**) TEM image, (**f–h**) Aberration-corrected HAADF-STEM image and the

corresponding intensity profiles in the (**g**) yellow square and (**h**) red square, (**i, j**) EDS element mapping, (**k**) XPS results of 41.8 wt% Pt SACs/PCN. **l** Pt *L*-edge XANES spectra, (**m**) Pt *L*-edge FT EXAFS spectra, and (**n**) the corresponding wavelet transformation results.

equipments is highly attractive for the practical preparation and application of high-loading SACs.

Herein, we report a general negative pressure annealing approach to construct UHL-SACs libraries, consisting of 13 different transition metal SACs supported on polymeric carbon nitride (PCN) with ultra-high metal loading. Based on the XAFS and in-situ aberration-corrected HAADF-STEM (AC HAADF-STEM), the negative pressure and thermal treatment enable fast dispersion of metal atoms over the support rather than aggregation, which are furtherly trapped by nitrogen sites (Fig. 1b). As a result, 13 different SACs are successfully prepared on PCN with ultra-high metal loadings of 27.3–44.8 wt%. Moreover, SACs and high-entropy single atoms (HESAs) composed of multiple metal sites with high metal loadings can be also readily obtained on N-doped carbon (NC). These evidences strongly validate the universality and scalability of the negative pressure annealing strategy for constructing a wide range of low-cost and high-areal-density SACs.

## Results and discussion

### Synthesis and structure investigation of Pt UHL-SACs

Given the wide applications of Pt-based catalysts[33], UHL-SACs of Pt are initially investigated. The PCN is used as the substrate firstly, which exhibits characteristic diffraction peaks at 2θ of 27.6 and 60.0° (Supplementary Fig. 1). The C 1s and N 1s XPS spectra provide additional confirmation of the formation of C-N bonding (Supplementary Fig. 2) with a N/C atomic ratio of 1.05, establishing ample coordinate nodes for UHL-SAC fabrication. The platinum-based UHL-SACs, denoted as Pt SACs/PCN, were synthesized by impregnating chloroplatinic acid onto PCN and subsequent annealing in a vacuum. To elucidate the crucial role of the negative pressure environment, a reference sample was prepared by annealing in Ar flow at 101 KPa (Pt NPs/PCN). The annealing pressure shows negligible impact on the apparent morphology and chemical constitution (Supplementary Figs. 3–5) of Pt-based catalysts compared with the PCN substrate. Moreover, the color of these two samples turns black from yellow following Pt deposition (Supplementary Fig. 6). Based on ICP analysis, the Pt contents are measured as 41.8 wt% and 40.9 wt% for Pt SACs/PCN and Pt NPs/PCN, respectively (Supplementary Table 2).

The TEM image of the Pt NPs/PCN (Fig. 1c) illustrates an accumulation of the metal particles with sizes around 10–50 nm. XRD pattern of Pt NPs/PCN (Fig. 1d) showcases the distinctive diffraction peaks characteristic of crystalline Pt, revealing the formation of Pt particles during annealing under ambient pressure. On the contrary, an absence of discernible diffraction peaks related to crystalline Pt was observed for Pt SACs/PCN. Additionally, the TEM image of Pt SACs/PCN (Fig. 1e and Supplementary Fig. 7) reveals no observable Pt particles, underscoring the effective prevention of metal aggregation under vacuum annealing conditions. To delve into the local structure of Pt sites in Pt SACs/PCN, an AC HAADF-STEM measurement was employed. Figure 1f, g reveal dense bright spots assigned to isolated Pt atoms are uniformly distributed over PCN, affirming the atomic dispersion of Pt sites on PCN randomly (Fig. 1h). Moreover, the average areal density of isolated Pt is estimated to 6.5 atoms/nm² based on the measured BET surface area of the PCN (Supplementary Figs. 8, 9), similar with the pixel statistics of Fig. 1g (5.6 atoms/nm²). These results identify the Pt SACs/PCN as one of the catalysts with the highest density of isolated Pt sites. Furthermore, EDS element mapping (Fig. 1i–j) affirm the uniform distribution of Pt, reinforcing the accuracy of these statistical findings. Figure 1k presents the XPS results of Pt SACs/PCN, which reaffirms the ultra-high Pt loading (survey) with a positive oxidation state evidenced by the Pt $4f_{7/2}$ binding energy of 72.7 eV. The XANES spectrum of Pt SACs/PCN, positioning the white line intensity between Pt foil and PtO₂, further verifies the partially positive oxidation state of Pt (Fig. 1l). The FT EXAFS spectra (Fig. 1m) are exploited to elucidate the coordination environment of Pt sites. The Pt SACs/PCN exhibits a dominant peak assigned to Pt-N

coordination at 1.6 Å, with the absence of Pt-Pt coordination at 2.5 Å. This result aligns seamlessly with wavelet transformation results (Fig. 1n), identifying the N-coordinated single-atom Pt sites in Pt SACs/PCN. These findings serve as conclusive evidence for successfully fabricating ultra-high loading single-atom catalysts via the negative pressure annealing approach.

To shed light on the formation of Pt SACs/PCN, the structure evolution of Pt species during the annealing process was investigated by temperature-dependent in-situ AC HAADF-STEM, ex-situ XAFS, and XPS, in vacuum and Ar condition respectively. The temperature-dependent in-situ AC HAADF-STEM images in vacuum conditions are shown in Fig. 2a and Supplementary Fig. 10. Dense bright spots are observed at 20 °C, revealing the uniform distribution of Pt precursor on the substrate. Moreover, no clusters and particles are generated along the temperature increasing from 20 to 400 °C, even after the sample is kept at 400 °C for 359 s. On the contrary, when annealing in the Ar flow, the atomic Pt can only be stabled below 300 °C (Fig. 2b). Observable Pt particles are generated when the temperature reaches to 300 °C. These particles grow bigger when the temperature is further increased to 400 °C. The corresponding EDS element measurements at different temperatures show a faster increase of Pt/Cl atomic ratio in vacuum than in Ar (Supplementary Table 3), demonstrating the accelerated Cl removal from Pt precursor in negative pressure conditions.

The coordination changes are investigated by XAFS. Figure 2c shows the Pt L-edge FT EXAFS spectra in vacuum conditions at different temperatures. Pt-Cl coordination is detected at 200 °C, which transfers to Pt-N coordination at 300 °C. Moreover, no Pt-Pt coordination is detected even at 400 °C, excluding the formation of Pt-Pt bonding. For samples annealed under Ar environment (Fig. 2d), the dominant Pt-Cl coordination at 200 °C is significantly decreased at 300 °C, companying with the formation of Pt-Pt coordination, which further takes the predominance at 400 °C. The FT EXAFS results are in good agreement with the in-situ AC HAADF-STEM and EDS results, demonstrating the different transformation pathways of Pt precursor in vacuum and Ar environment. The oxidation states of Pt under different annealing pressures show down-hill tendencies along the increasing temperature (Fig. 2e, f), which may due to the loss of Pt-Cl bonding and the formation of Pt-N or Pt-Pt coordination. However, for the sample annealed in Ar, the Pt oxidation state is lower than that annealed in vacuum (Fig. 2g). This can be attributed to the weaker electronegativity of Pt than N. These results illustrate the evolution of Pt UHL-SACs, that is Pt-Cl coordination rapidly dissociates at relatively low temperatures to generate active Pt species, and the vacuum condition greatly suppresses the metal aggregation via promoting the Pt-N coordination at relatively high temperatures, thus enables the formation of high-density Pt SACs.

### Universally preparing metal UHL-SACs

To reveal the generality of the negative pressure annealing approach, this synthetic process is extended to 12 other single-atom metal sites on PCN (M SACs/PCN, M = V, Cr, Mn, Fe, Co, Ni, Cu, Zn, Nb, Mo, Ir and Au, Fig. 3a). The metal loadings of all these catalysts are measured between 27.3 wt% to 44.8 wt%, and the metal areal density are confirmed at a high level (Fig. 3b, Supplementary Table 2, and Supplementary Fig. 9). The white-line intensities from the XANES spectra indicate the positive oxidation state of the metal sites in M SACs/PCN (Supplementary Fig. 11). The AC HAADF-STEM (Fig. 3c) and FT-EXAFS spectra (Fig. 3d) identify the N-coordinated single-atom metal sites, and the absence of the metal-metal coordination excludes the formation of the metal clusters. The characterizations of XRD, TEM, EDS mapping, and XPS are shown in Supplementary Figs. 12–24, which further indicate the uniform distribution of the positive-charged isolated metal sites on the PCN substrate, and no aggregation of the metal is detected.

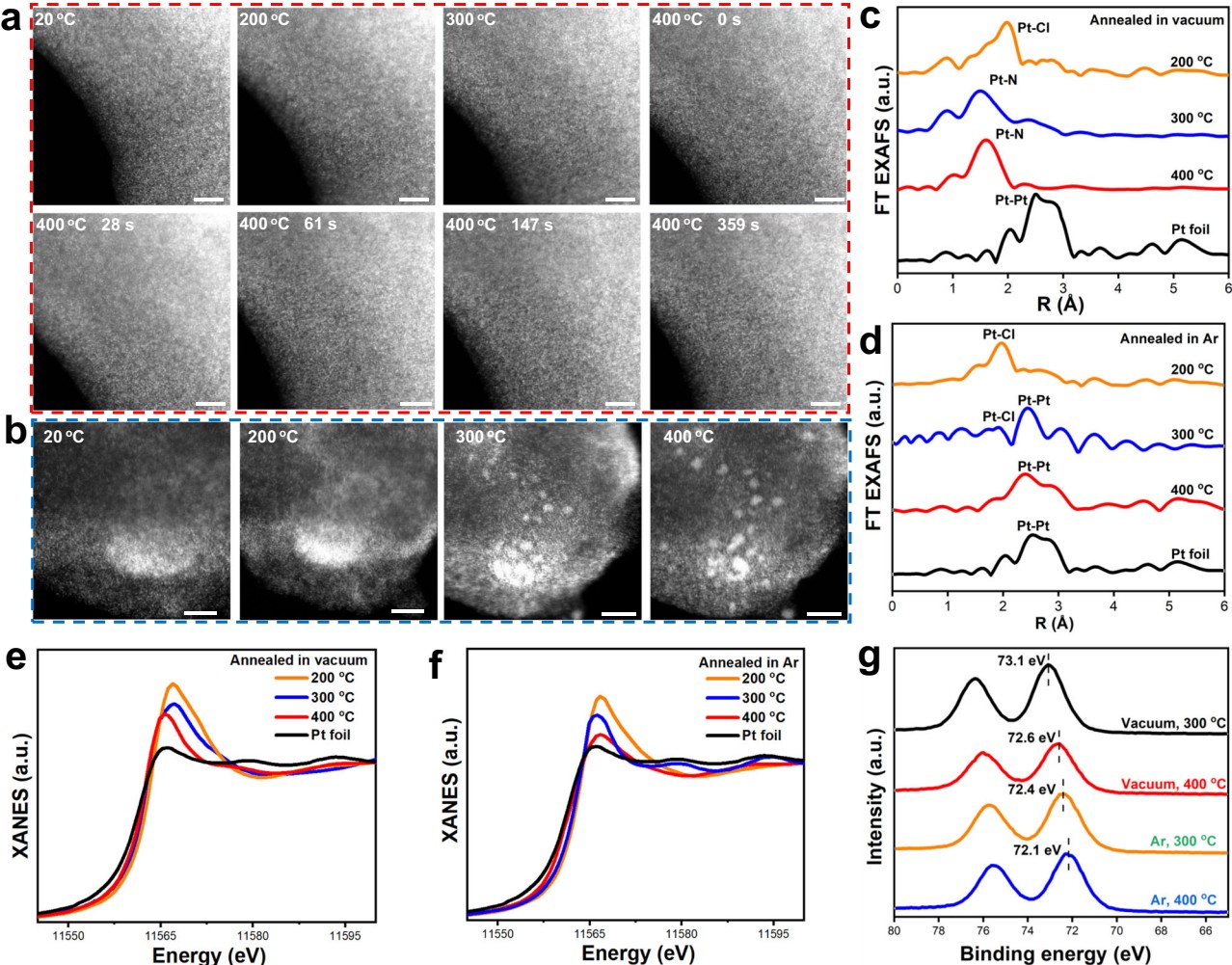

**Fig. 2 | Insight of the Pt species transformation during the generation of Pt SACs/PCN and Pt NPs/PCN.** Temperature-dependent in-situ aberration-corrected HAADF-STEM images of (**a**) Pt SACs/PCN and (**b**) Pt NPs/PCN, the scale bar represents 5 nm. **c**, **d** Pt L-edge FT EXAFS spectra and (**e**, **f**) XANES spectra of samples annealed at different temperatures in (**c**, **e**) vacuum and (**d**, **f**) Ar. **g** Pt 4 f XPS spectra of samples annealed at different temperatures in vacuum and Ar.

The adaptability of the negative pressure annealing approach on different substrates is also investigated. The NC obtained via the pyrolysis of guanine is used instead of PCN to prepare UHL-SACs (M SACs/NC, M=Pt, Fe, Co, Ni, and Cu). The XRD of the NC substrate reveals the structure of graphitic carbon (Supplementary Fig. 25). The XPS results confirm the doping of N on carbon (Supplementary Fig. 26). The characterizations, including AC HAADF-STEM, FT-EXAFS, EDS element mapping, XANES, XRD, and XPS (Fig. 4a–e and Supplementary Figs. 27–34), identify the N-coordinated single-atom metal sites with positive oxidation states in these M SACs/NC. The mental contents are measured as high as 34.1 wt% (Pt), 21.5 wt% (Fe), 19.6 wt% (Co), 17.7 wt% (Ni), and 29.8 wt% (Cu) (Supplementary Table 2), demonstrating the obtain of UHL-SACs with high metal areal density on NC substrate (Supplementary Figs. 35, 36). Moreover, high-entropy single atoms (HESACs) containing Pt, Fe, Co, Ni, and Cu are also prepared on the NC. As shown in Fig. 4f, g and Supplementary Fig. 37, all five metals distribute uniformly on the NC as N-coordinated isolate metal sites. The metal contents are 15.6, 3.1, 4.1, 2.3, and 7.3 wt% for Pt, Fe, Co, Ni, and Cu, respectively, resulting in an overall metal content of 32.4 wt% (Fig. 4h, Supplementary Table 2). The positive oxidation states of the metal sites are confirmed (Supplementary Figs. 38, 39). Furthermore, no condensed matter of any metal is detected (Fig. 4i and Supplementary Figs. 40, 41). Although a limited number of metals were tested

on NC, we speculate that the formation of SACs on NC follows the same evolution pathway as that on PCN, demonstrating the versatility of this synthetic method on different N-containing carbon substrates for preparing SACs with multiple metals. Together, this work provides a universal synthetic strategy to fabricate various UHL-SACs and even HESACs.

## Catalytic evaluation of Pt SACs/PCN

The partial oxidation of propane to valuable liquid oxygenates represents a novel strategy to utilize this class of light alkane[34]. Among the several current strategies, such as electrocatalysis[35], photocatalysis[36,37], thermal-derived homogeneous[38], and thermal-derived heterogeneous catalysis, the exploitation of heterogeneous catalyst shows the greatest application potential[39–41]. However, the consumption of costly oxidants poses an obstacle to it. To address this issue, a catalytic process that can transfer propane to oxygenates with low-cost oxidants is urgently needed. Inspired by the molecular oxygen activation capacity of isolated Pt sites[2], Pt SACs/PCN is evaluated in the oxidation of propane with oxygen in this work.

The reaction is performed at 175 °C in a 240 mL autoclave, with propane (5 bar) as reactant, oxygen (6 bar) as oxidant, and acetonitrile (70 ml) as solvent (Fig. 5a). The decreased pressure and the gas chromatography (GC) results indicate the consumption of propane, and

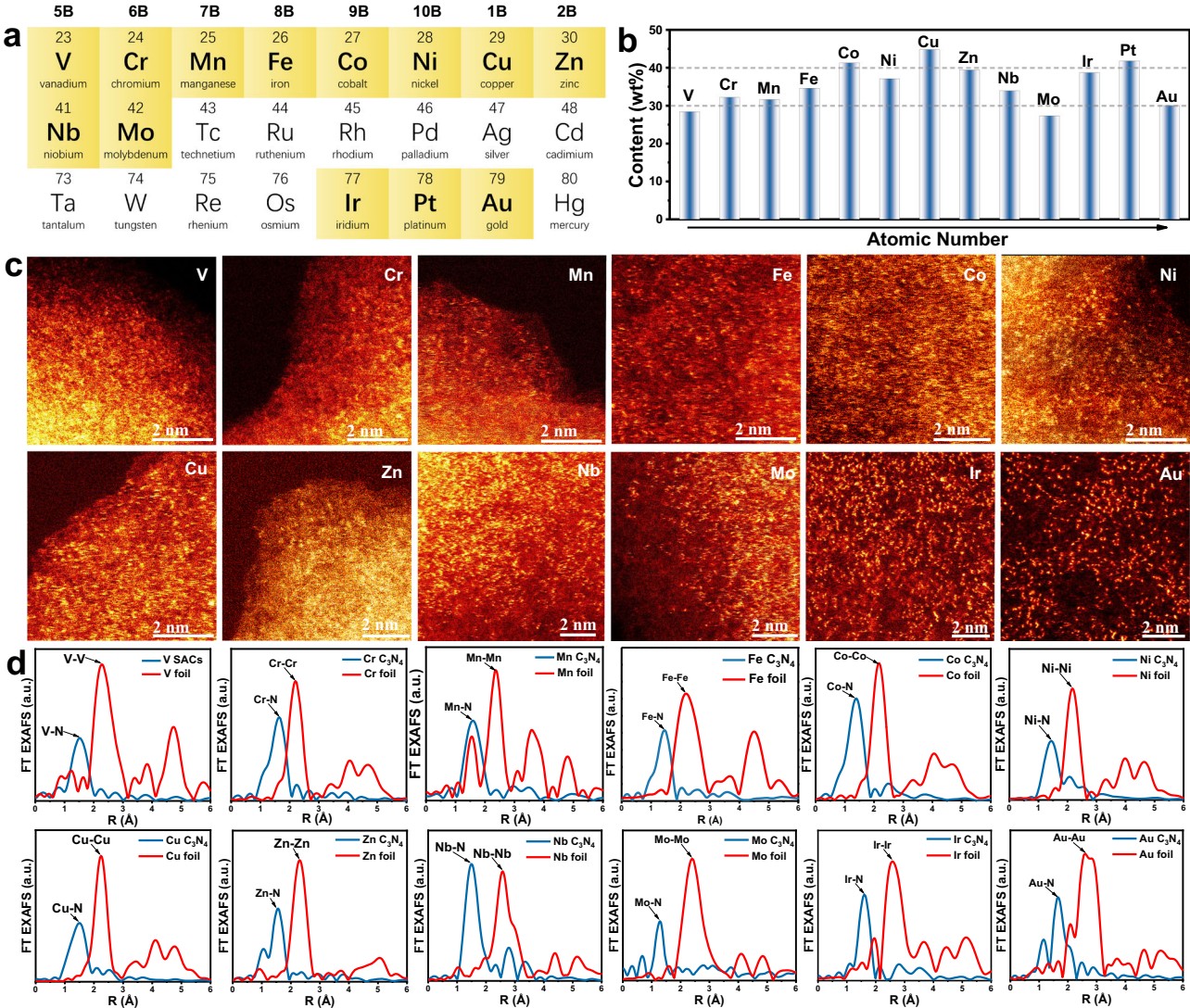

**Fig. 3 | The universal preparation of UHL-SACs on PCN (M SACs/PCN). a** The metal elements used for fabricating M SACs/PCN. **b** The metal content in the as-prepared catalysts. **c** Aberration-corrected HAADF-STEM and (**d**) FT-EXAFS spectra of various M SACs/PCN.

the gas production is identified as CO (Supplementary Fig. 42). Interestingly, liquid productions are detected in acetonitrile, which is dominated by oxygenates (acetone, acetic, and methanol, et al.), revealing the capacity of Pt SACs/PCN in transforming propane into valuable liquid productions (Fig. 5b, c and Supplementary Fig. 43). These liquid productions are further quantitatively analyzed via the external standard method (Supplementary Figs. 44, 45). As shown in Fig. 5d, the liquid product is confirmed as 37.1 mmol/g$_{cat}$ at 3 h, which increases with the reaction time, and reaches 71.9 and 107.6 mmol/g$_{cat}$ at 6 and 9 h, surpassing the low-loading Pt SACs/PCN, Pt nanoparticles (Pt NPs/PCN) and commercial Pt/C catalyst (Fig. 5e). To reveal the intrinsic activity of Pt SACs/PCN, the turnover frequency (TOF) and mass-specific activity are confirmed as $1.6 \times 10^{-3}$ mol$_{pro}$·mol$_{Pt}^{-1}$·s$^{-1}$ and 12.0 mmol/g$_{cat}$/h (Fig. 5f, g and Supplementary Table 4), well-placed among select prior reports of propane activation performance with oxygen. Interestingly, among the catalysts working with oxygen, only Pt SACs/PCN selects the pathway toward oxygenates (Fig. 5g). This may be the first observation of heterogeneously catalytic oxidation of propane to oxygenates with oxygen, which provides a strategy to utilize propane for harvesting valuable liquid productions. Moreover, the catalytic performance of Pt SACs/PCN shows insignificant decay after be reused five times (Fig. 5h), and the used catalyst maintains the

dense isolated Pt sites (Supplementary Fig. 46), confirming its stability. To clarify the effect of the substrates, Pt SACs on NC were also evaluated in the propane oxidation (Fig. 5e). Following a similar trend with Pt SACs/PCN, Pt SACs/NC with higher Pt loading shows better activity than those with lower Pt loading and Pt particles, and the productions are dominated by oxygenates. These catalyst evaluations demonstrate the potential application of high-loading Pt SACs in activating light alkanes.

In conclusion, we report a general negative pressure annealing strategy to fabricate ultrahigh-loading single-atom catalysts across a broad range of transition metals. Besides monometallic SACs, high-entropy single-atom catalysts that contain multiple metal single atoms with high metal contents can also be obtained, proving the general applicability of the pressure annealing method. In-situ microscopic studies combined with ex-situ XAFS reveal the pivotal role of the vacuum annealing condition in suppressing the aggregation of metal species, enabling the formation of dense N-coordinated Pt sites. Furthermore, UHL Pt SACs/PCN exhibits superior catalytic performance in the oxidation of propane towards valuable liquid production. These findings provide valuable guidance for preparing a wide range of high-density SACs and show great potential use in efficient catalytic transformations.

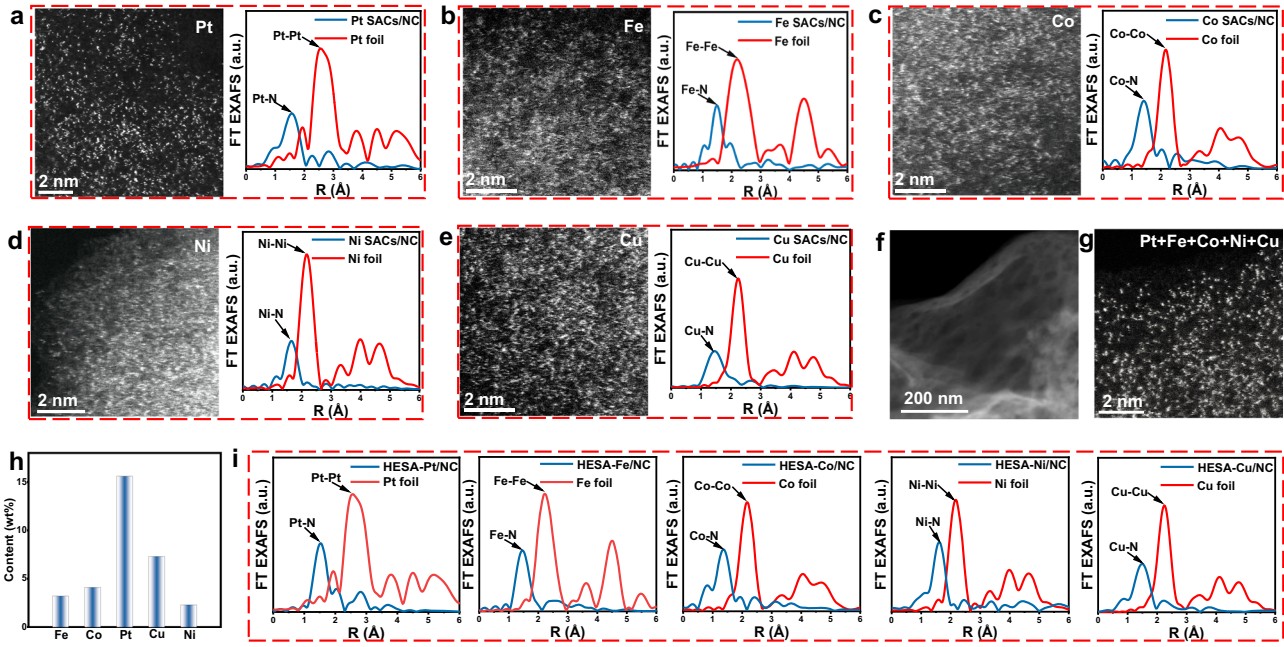

**Fig. 4 | Fabricating UHL-SACs on N-doped Carbon (NC) substrate.** The aberration-corrected HAADF-STEM images and FT-EXAFS spectra of NC supported UHL-SACs with (**a**) Pt, (**b**) Fe, (**c**) Co, (**d**) Ni, and (**e**) Cu. **f** TEM and (**g**) Aberration- corrected HAADF-STEM images of the UHL HESACs. **h** The metal content in the HESACs. **i** FT-EXAFS spectra for the metals in HESACs.

## Methods

**Materials.** Guanine, melamine, vanadyl sulfate, chromic nitrate, man- ganous nitrate, ferric nitrate, cobaltous nitrate, nickel nitrate, cupric nitrate, zinc nitrate, niobium oxalate were purchased from Sinopharm Chemical Reagent Co. Molybdenum pentachloride, chloro-iridic acid, chloroplatinic acid, tetrachloro-auric acid were purchased from Alad- din. All solutions were prepared using deionized water. All the che- micals were used without further purification.

**Synthesis of PCN.** The mixture of melamine and dicyandiamide (molar ratio = 7:3) was pyrolyzed in a tube furnace at 600 °C for 1.5 h (Ar flow, 100 sccm). The obtained powder (3 g) was treated in 65 wt% HNO₃ (50 ml) at 80 °C for 6 h, followed by an ultrasonic treatment for 1 h. The suspension was centrifuged and washed with deionized water to gain yellow powder, which was labeled as PCN.

**Synthesis of NC.** Guanine was pyrolyzed in a tube furnace at 600 °C for 1.5 h (Ar flow, 100 sccm). The obtained black powder was labeled as N-doped carbon (NC).

**Synthesis of Pt SACs/PCN, Pt NPs/PCN, and Pt SACs/NC.** Pt SACs/ PCN were prepared via the impregnation-vacuum pyrolysis method. 20 ml chloroplatinic acid solution (0.075 mol/L) was added in 1 g PCN. The obtained suspension was ultrasonic treated for 15 min, aged at room temperature for 2 h, and dried at 80 °C for 8 h. The obtained powder was placed in a tube furnace. The tube furnace was firstly purged by 100 sccm Ar flow for 15 min. Then, the outlet was connected to an operating mechanical pump (limiting pressure 6 × 10⁻² Pa), and the inlet was closed. After 1 h of pyrolysis under vacuum conditions at 400 °C, the black powder Pt SACs/PCN was obtained. Pt/NPs/PCN was prepared via the same procedure, except for the pyrolysis process, which was carried out in Ar flow (atmospheric pressure). Pt SACs/NC was prepared by the impregnation-vacuum pyrolysis method, using 0.035 mol/L chloroplatinic acid solution and NC substrate.

**Synthesis of M SACs/PCN and M SACs/NC.** M SACs/PCN (M = V, Cr, Mn, Fe, Co, Ni, Cu, Zn, Nb, Mo, Ir and Au) was synthesized via the same method with Pt SACs/PCN, using vanadyl sulfate, chromic nitrate, manganous nitrate, ferric nitrate, cobaltous nitrate, nickel nitrate, cupric nitrate, zinc nitrate, niobium oxalate, molybdenum pen- tachloride, chloro-iridic acid, and tetrachloro-auric acid as the metal

precursor, respectively. The concentration of metal precursor solution was 0.17 mol/L for V, Cr, Mn, 0.2 mol/L for Fe, Co, Ni, Cu, and Zn, 0.12 mol/L for Nb and Mo, 0.075 mol/L for Ir, and 0.045 mol/L for Au. The vacuum pyrolysis temperature for V, Cr, Mn, Fe, Co, Ni, Cu, and Zn SACs/PCN was 500 °C. M SACs/NC (M = Fe, Co, Ni, Cu) were prepared according to the method of M SACs/PCN. The concentration of the metal precursor solution was 0.07–0.1 mol/L.

**Synthesis of HESACs.** HESACs were prepared according to the method of M SACs/NC. The metal precursor solution was a mixture of ferric nitrate, cobaltous nitrate, nickel nitrate, cupric nitrate, and chloroplatinic acid. Their concentration in the solution was 0.015, 0.015, 0.01, 0.03, and 0.02 mol/L, respectively.

**Characterization.** The transmission electron microscopy (TEM) images and the corresponding element mapping were recorded on a JEOL-2100F FETEM. The electron acceleration energy was 200 kV. The morphologies of the samples were measured on Thermo-Fisher Apreo S scanning electron microscope. HAADF-STEM images were recorded on the JEOL JEMARM200F TEM/STEM system. The in-situ heating aberration-corrected high-angle annular dark field scanning transmis- sion electron microscope (Cs-HAADF-STEM) experiments were con- ducted on a double-spherical aberration corrected FEI Titan Themis Z scanning transmission electron microscope with an accelerating vol- tage of 300 kV combined with the in-situ heating holder provided by FEI company. The prepared precursor was dispersed onto the in-situ heating chip purchased from FEI company. Before STEM imaging, beam shower was carried out for 15 min. A probe size of 9 was used to image. The heating rate was set as 10 °C/min. To exclude the influence of e-beam irradiation, the images of control areas were collected only at each temperature point. The crystallographic phase of the as- synthesized materials was characterized by X-ray powder dif- fractometer (XRD, Bruker D8 Advance, λ = 1.5418 Å). Raman spectro- scopy (WITec alpha300R, 532 nm laser) was used to study the structure and disorder of the catalysts. X-ray photoelectron spectro- scopy (XPS) measurements were performed on a PHI 5000 VersaProbe III X-ray photoelectron spectrometer using an Al Kα X-ray source. The power is 12.5 W, the bandpass energy of the full spectrum test is 280 eV, and the bandpass energy of the fine spectrum test is 112 eV.

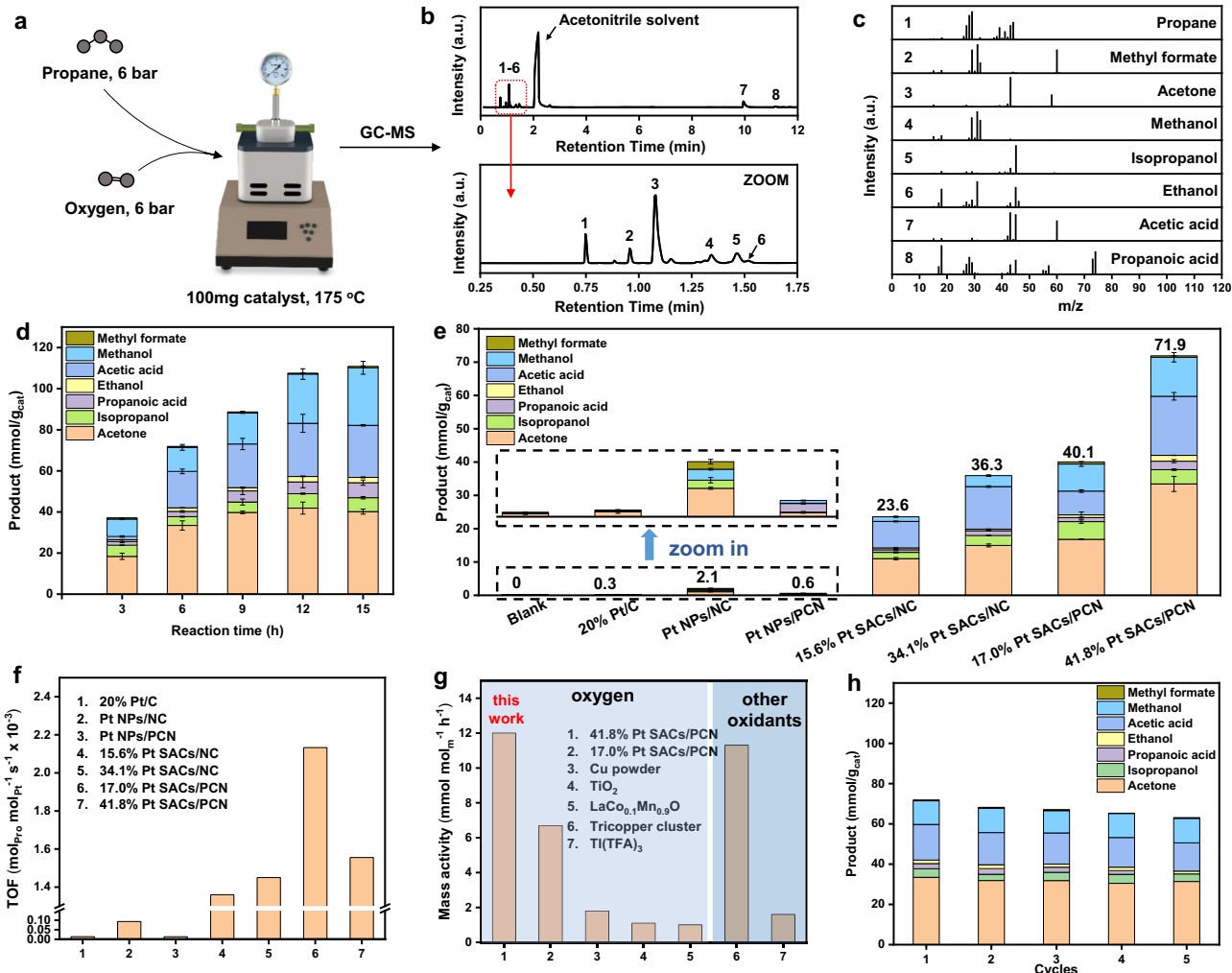

**Fig. 5 | Catalytic evaluation of Pt SACs/PCN in propane oxidation. a** Illustration of the reaction condition. **b** Gas chromatographic profile and (**c**) mass spectra of the liquid productions measured by GC-MS. **d** Catalytic performance of Pt SACs/PCN over reaction time. **e** Catalytic performance of various catalysts at 6 h. **f** TOF value of the catalyst used in this work. **g** Comparison of the propane oxidation performance with previous reports. **h** Stability test. The maximum measurement error for (**d**, **e**) is ±3.8%, which represent the standard deviation of 3 replicates at least.

The spectra are calibrated by C1s (284.8 eV), and the spectral analysis is performed by Avantage software. Inductively coupled plasma atomic emission spectroscopy (ICP-AES) was employed with PerKinElmer Optima 2100DV to detect the loading content of metal on catalysts. X-ray Absorption Fine Structure Spectroscopy (XAFS) were collected at beamline 1W1B of Beijing Synchrotron radiation Facility and beamline BL14W1 of Shanghai Synchrotron Radiation Facility. The acquired XAFS data were analyzed by Athena and Artemis software modules in IFEFFIT software package.

Catalytic performance test. The catalyzed oxidation of propane with molecular oxygen was carried out in a 240 mL autoclave reactor. In a typical run, 70 ml acetonitrile solvent and 100 mg catalyst were added in the reactor. And, 5 bar propane and 6 bar oxygen was then added in the reactor According to the ideal gas equation, the added propane was about 35.4 mmol. The reaction was carried out at 175 °C. After the reaction, the reactor was cooled to room temperature. The liquid phase was flited and analyzed by a gas chromatograph-mass spectrometer (Agilent 8890-5977B). The qualitative analysis was performed on a HP-5 MS capillary column (inner diameter 0.25 μm, length 30 m), and the quantitative analysis was performed with FID detector (as shown in Supplementary Figs. 43, 44), using HP-5 capillary column (0.25 μm, 30 m) and calculated by the external standard method

(Supplementary Fig. 45). The gas phase was collected and analyzed by a Panna A60 gas chromatography. The yield of liquid oxy-compounds after 9 h reaction over Pt SACs/PCN is calculated by the moles of carbon in liquid oxy-compounds (23.19 mmol carbon) and the carbon inlet (35.4 mmol propane, 106.2 mmol carbon).

## Data availability

The data that support the findings of this study are available within the article and its Supplementary Information files. All other relevant data supporting the findings of this study are available from the corresponding authors upon request.

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

## Acknowledgements

The authors thank the photoemission endstation beamline 1W1B station in the Beijing Synchrotron Radiation Facility (BSRF) and beamline BL13SSW, BL14W1 in Shanghai Synchrotron Radiation Facility (SSRF) for help with the XAFS characterizations. This work is financially supported by the National Natural Science Foundation of China (22275147 to Y.Q., 52301289 to K.L., 21902150 to Y.Q., 52122212, 12274391 and 22321001 to Y. L.), Natural Science Basic Research Program of Shaanxi (2022JM-018 to Y.W. and 2022JQ-082 to K.L.), the Youth Innovation Promotion Association of CAS (2020458 to Y. L.), the Key Research Program of Frontier Sciences, CAS (ZDBS-LY-SLH003 to Y. L.).

## Author contributions

Y.W., C.L., and X.H. contribute to this work equally. Y.W. designed the basic principle and arranged the experiment. C.L. experimented. Y.Q. and K.L. worked together on the characterizations and article writing. Jinbo. B. funded this work. L.Z. and C.H. performed the XAFS measurement. X.H. and Y.L. performed the in-situ aberration-corrected HAADF-STEM. Jintao. B., Z.L., and X.W. revised the writing work.

## Competing interests

The authors declare no competing interests.
