## [Peer Review File · Nature Communications]

REVIEWER COMMENTS

Reviewer #1 (Remarks to the Author):

In this work the authors report a remarkable confluence of three advances: 1) several simple to make, nitrogen-laden carbon supports, 2) a simple, potentially easily scalable, and generalizable method to apply metals to them which results in atom isolation at notably high areal and mass loadings, and 3) an apparently new reaction for which only the isolated form of the catalytic material is active, in effect, heterogenizing a homogeneous catalyst.

Before getting too far, the comment should be made that the quality of the English is inversely proportional to the apparently superb quality of the science described. There are incorrect terms (“inoperando” was used in place of “operando” and in fact, “operando” should not have been used at all, but rather “in-situ”, “structure revolution” should read “structural evolution,” etc.), sentence fragments (e.g. 2nd full sentence on p. 3), misspellings (“dopped” instead of “doped”), improper verb tenses, articles, and plurals. The text demands a major scrubbing, but one well worth the effort for the very nice science presented here.

Perhaps the only factor which limits the impact of the work is that the method appears to be applicable only to a single class of catalyst supports, nitrogen-doped carbons. With these materials, however, the density of isolated atoms achievable, above 5 atoms/nm², appears to be among the highest now possible.

The strength of the paper is the comprehensive characterization of the materials (by XRD, STEM and TEM, XPS, and XAS). The suite of techniques employed renders a comprehensive, consistent picture of the final materials. The comparison with control samples comprising nanoparticles adds to the soundness of the characterization.

I can offer several suggestions on how to improve the solidity of the science and so maximize the impact of the paper.

1) In the introduction, focus not only on catalyst weight loadings, but on areal loadings. Can Figure 1a be recast in terms of atoms/nm²? That might make the current results stand out even more.

2) Perhaps the one weakness in the characterization is the absence of BET surface area of the carbon materials. Having this value would allow a robust calculation of the areal loadings, instead of the

estimate from the “pixel statistics” in Figure 1. BET and pore size distributions of the PCN and NC material are really essential here.

3) Relatively little attention has been paid to the reaction in light of the literature. How novel is this material for this reaction? How novel is this reaction? Are there any comparisons which can be made? Can the reaction rates be cast in terms of turnover frequencies (molecules propane reacted per Pt atom per second) and compared with other catalysts in the literature? Has this reaction previously been possible only with a homogeneous catalyst, in which case the results would be even more important.

4) Why was the reactivity of NC series of catalysts not evaluated? Showing a similar trend of activity would further boost the impact.

5) Given that the results of Figure 5e are central to the science, it is essential that these results are replicable. Has this been done? If not, it should be, and error bars should be added to the figure or at least commented on in the text.

In sum, getting beyond the poor English, this paper was a thrill to read. The work appears to be a wonderful advance in synthesis science, combined with a new class of nitrogen-doped carbons to allow atom isolation at previously unachievable areal loadings, and which show high activity for pertinent catalytic reactions. It should not be difficult to tighten up the loose ends in the manner suggested.

Reviewer #2 (Remarks to the Author):

The authors designed a generalized negative pressure annealing method for the preparation of low-cost, high-density, ultra-high-loaded single-atom catalysts (SACs) on carbon nitride substrates. Negative pressure and heat treatment lead to rapid dispersion rather than aggregation of metal atoms, which are subsequently captured by nitrogen sites, and ultrahigh metal loadings ranging from 27.3% to 44.8% can be achieved for multiple SACs on polymerized carbon nitride. In addition, high-entropy SACs with high metal content containing multiple metal single atoms have been obtained on carbon nitride, demonstrating the general applicability of the negative pressure annealing method. I would recommend this manuscript to be accepted with some revisions as mentioned in the following.

1. Please describe in detail the process of synthesizing Pt UHL-SACs by pyrolysis.

2. For discussing the adaptability of the negative pressure annealing method on different substrates, the authors have selected only five metals for single-atom doping. Whether the versatility of this method can be proved, please explain it.

3. The catalytic evaluation of Pt SACs/PCNs in Chapter 2.3 is not comprehensive enough, and more discussion is needed to demonstrate the excellent performance of SACs.

4. Some classic or recent references about SACs also need to be added, such as *Nature*, 2023, (622):754-760, *Nano Energy*, 2023, (111): 108404, *Sci. China Mater.*, 2023, 66(3): 1080-1089, and *J Mater. Chem. A*, 2022, (10): 6231-6241.

5. Details need to be adjusted, e.g., N-doped Carbon should be annotated with the full name on the first occurrence, and the same phrase should be replaced by an acronym on subsequent occurrences. The quality of the pictures is not high enough, and the resolution of the graphs should be increased (such as Fig. 1 and Fig. 3). Too many pictures are put in Fig. 4, resulting in the pictures not being clear enough. It is recommended that Fig. 4 be readjusted.

Responses to Reviewers:

The reviewers' comments are laid out below in *Italic font* and specific concerns have been numbered. Our responses are given in normal font and changes/additions to the manuscript and supplementary information are highlighted by using **red colored text**.

Reviewer #1 (Remarks to the Author):

In this work the authors report a remarkable confluence of three advances: 1) several simple to make, nitrogen-laden carbon supports, 2) a simple, potentially easily scalable, and generalizable method to apply metals to them which results in atom isolation at notably high areal and mass loadings, and 3) an apparently new reaction for which only the isolated form of the catalytic material is active, in effect, heterogenizing a homogeneous catalyst.

Before getting too far, the comment should be made that the quality of the English is inversely proportional to the apparently superb quality of the science described. There are incorrect terms (“inoperando” was used in place of “operando” and in fact, “operando” should not have been used at all, but rather “in-situ”, “structure revolution” should read “structural evolution,” etc.), sentence fragments (e.g. 2nd full sentence on p. 3), misspellings (“dopped” instead of “doped”), improper verb tenses, articles, and plurals. The text demands a major scrubbing, but one well worth the effort for the very nice science presented here.

Answer: Thanks, we had checked the manuscript carefully to improve the quality of the English in the revised manuscript, including incorrect terms, sentence fragments, misspellings, improper verb tenses, articles, and plurals.

Perhaps the only factor which limits the impact of the work is that the method appears to be applicable only to a single class of catalyst supports, nitrogen-doped carbons. With these materials, however, the density of isolated atoms achievable, above 5 atoms/nm², appears to be among the highest now possible.

The strength of the paper is the comprehensive characterization of the materials (by XRD, STEM and TEM, XPS, and XAS). The suite of techniques employed renders a comprehensive, consistent picture of the final materials. The comparison with control samples comprising nanoparticles adds to the soundness of the characterization.

I can offer several suggestions on how to improve the solidity of the science and so maximize the impact of the paper.

1. In the introduction, focus not only on catalyst weight loadings, but on areal loadings. Can Figure 1a be recast in terms of atoms/nm²? That might make the current results stand out even more.

Answer: Thank you very much for highlighting the importance of the metal areal loading for our work. We tried to recast Figure 1a in terms of atoms/nm², but most of the literatures cited in Figure 1a did not provide enough information, such as the BET surface area of the carriers, to calculate the areal density of the Pt SACs. To address this issue, we analyzed the currently available works which presented enough information to evaluate the Pt areal density. The related results were listed in Figure S1, and the pivotal role of the areal loadings was discussed along with the weight loadings in the Introduction section in the revised manuscript.

Table S1. Pt areal loading of reported Pt SACs.

	Sample	Pt areal density (atoms/nm ²)	Ref.
1	Pt ₁ /PCN	4.1	Nat. Nanotech., 2022, 17 , 174-181
2	Pt ₁ /CeO ₂	3.9	Angew. Chem. Int. Ed., 2022, 61 , e202212338
3	Pt/CeO ₂	<1.2	Catal. Today 2024, 425 , 114298
4	Pt ₁ /Fe ₂ O ₃	1.2	Nanotechnology, 2018, 29 , 204002
5	Pt(0.25)/TiO ₂	0.1	Nature Commun., 2024, 15 , 998
6	Pt ₁ -N/BP	0.01	Nature Communications, 2017, 8 , 15938
7	Pt _{1.1} /BP _{defect}	0.02	Angewandte Chemie, 2018, 131 , 4, 1175
8	S-Pt-C ₃ N ₄	<0.5	Angewandte Chemie, 2020, 132 , 15, 6283
9	Pt ₁ -CN	<0.5	Advanced Materials, 2016, 28 , 12, 2427
10	Pt ₁ /Fe ₂ O ₃	<0.5	Nature Communications, 2019, 10 , 4500
11	Pt ₁ /Al ₂ O ₃	0.9	Science Advances, 2020, 6 , 25
12	Pt ₁ /NC	<0.5	Nature Communications, 2019, 10 , 1278
13	PtSA-MNSs	0.6	Angewandte Chemie, 2019, 58 , 30, 10198
14	20Pt/meso S-C	0.9	Science Advances, 2019, 5 , 10
15	Pt SACs	<0.5	Nature Communications, 2019, 10 , 4585

Page 3, line 62-73 “Nevertheless, considering their high surface energy, the SACs are generally constructed with low metal loadings to circumvent the aggregation of metal atoms into metal clusters or nanoparticles¹⁷. This results in a low metal areal density. Taking the Pt SACs as a typical example, they display impressive activity and

selectivity in the thermal-driven activation of light alkane¹⁸⁻²⁰. A literature analysis shows that the metal contents of most Pt SACs are 2 wt% or below (Figure. 1a)²¹⁻²⁴, and the Pt areal density of most catalysts are hard to surpass 1.5 atoms/nm² (Table S1). In this case, the SACs with insufficient areal density of active sites not only limit their overall catalytic performance but also decrease the productivity per unit volume or mass of catalysts. Therefore, the development of a universal synthesis strategy for accessing SACs with high metal loading and sufficient areal density is significant in this field, but yet challenging²⁵⁻²⁸.”

2. Perhaps the one weakness in the characterization is the absence of BET surface area of the carbon materials. Having this value would allow a robust calculation of the areal loadings, instead of the estimate from the “pixel statistics” in Figure 1. BET and pore size distributions of the PCN and NC material are really essential here.

Answer: Thanks to the professional comments. According to the suggestion of the reviewer, N₂ adsorption/desorption tests were measured and the results were presented in Figure S8 and S35. Accordingly, the BET surface area of PCN and NC were confirmed as 341.7 and 390.6 m²/g, respectively. Moreover, the metal areal loadings of various SACs were calculated according to the metal weight-loading and the BET surface area of the carbon materials. These results were showed in Figure S9 and S36, and discussed in the revised manuscript.

Page 6, line 135-138, “Moreover, the average areal density of isolated Pt is estimated as 6.5 atoms/nm² based on the measured BET surface area of the PCN (Figure S8-9), similar with the pixel statistics of Figure 1g (5.6 atoms/nm²). These results identify the Pt SACs/PCN as one of the catalysts with the highest density of isolated Pt sites.”

Page 8, line 196-198, “The metal loading of all these catalysts is measured between 27.3 wt% to 44.8 wt%, and the metal areal density are confirmed at a high level (Figure 3b, Table S2, and Figure S9).”

Page 9, line 218-221, “The mental content is measured as high as 34.1 wt% (Pt), 21.5 wt% (Fe), 19.6 wt% (Co), 17.7 wt% (Ni), and 29.8 wt% (Cu) (Table S2), demonstrating the obtain of UHL-SACs with high metal areal density on NC substrate (Figure S35-36).”

Figure S8. (a) N₂ adsorption/desorption isotherms with BET surface area and (b) pore size distribution of PCN.

Figure S9. Areal density of metal atoms in UHL-SACs based on PCN carrier. The areal density was estimated based on the bulk metal content and the specific surface area of the carrier, assuming that all of the metal remains on the carrier surface. Although wet deposition approaches are known to promote surface localization, the metals were predicted to percolate preferentially into the bulk of graphitic carbon nitride (*ACS Catal.* 2020, **10**, 11069, and *Nat. Nanotech.*, 2022, **17**, 174-181.), which might explain the relative high value.

Figure S35. (a) N₂ adsorption/desorption isotherms with BET surface area and (b) pore size distribution of NC.

Figure S36. Areal density of metal atoms in UHL-SACs based on NC carrier. The Areal density was estimated based on the bulk metal content and the specific surface area of the carrier, assuming that all of the metal remains on the carrier surface. Figure SX. Areal density of metal atoms in UHL-SACs based on PCN carrier. The Areal density was estimated based on the bulk metal content and the specific surface area of the carrier, assuming that all of the metal remains on the carrier surface. Although wet deposition approaches are known to promote surface localization, the metals were predicted to percolate preferentially into the bulk of graphitic carbon nitride (*ACS Catal.* 2020, **10**, 11069, and *Nat. Nanotech.*, 2022, **17**, 174-181.), which might explain the relative high value.

3. *Relatively little attention has been paid to the reaction in light of the literature. How novel is this material for this reaction? How novel is this reaction? Are there any comparisons which can be made? Can the reaction rates be cast in terms of turnover frequencies (molecules propane reacted per Pt atom per second) and compared with other catalysts in the literature? Has this reaction previously been possible only with a homogeneous catalyst, in which case the results would be even more important.*

Answer: Thank you for the insightful comments. Activation of propane with catalysts and oxidants represents a novel strategy to utilize this class of chemicals. This process can be roughly divided into two pathways based on the class of productions: one is the oxidative dehydrogenation of propane to gaseous propylene (*Nature Catal.*, 2023, **6**, 666-675); and the other is the partial oxidation of propane to liquid oxygenates. The latter gain liquid productions which are valuable and easy to store and transport, but

poses challenges on the design of catalysts and catalytic process to regulate the complex production distribution. Pioneering efforts had been made to develop catalytic process for propane oxidation to oxygenates, such as electrocatalysis (*J. Am. Chem. Soc.*, 2021, **143**, 3967-3974), photocatalysis (*Catal. Lett.*, 1997, 44, 247-253 and *Catal. Lett.*, 2013, **143**, 154-158) and thermal-derived homogeneous catalysis (*Science*, 2014, **343**, 1232-1237). Several heterogeneous catalysts were also investigated, which may be more attractive in industrial process than homogeneous catalyst (*ACS Sustainable Chem. Eng.*, 2018, **6**, 5431-5440, *Catal. Today*, 1999, **49**, 171-175 and *Catal. Sci. Technol.*, 2023, **13**, 4839-4846), but the consumption of costly oxidants hinders their further applications. Thus, the development of heterogeneous catalysts that can convert propane to oxygenates with costless oxidant, such as oxygen or air, is particularly important for the utilization of propane.

To address this issue, isolated Pt sites was used in this work as catalyst for propane oxidation to oxygenates, considering its capacity in activation molecular oxygen (*Nat. Chem.*, 2011, **3**, 634-641). Moreover, the catalytic performance of Pt SACs was promoted by increasing the loading of atomic Pt. By compared with the previous reports, our catalyst showed superior performance in activating propane with O₂, but chose a unique pathway towards oxygenates (**Figure 5f-g and Table S4**). Actually, this work may be the first investigation on the heterogeneous catalytic oxidation of propane to oxygenates with oxygen, which can provide a novel strategy to utilize propane for harvesting valuable liquid productions.

According to the discussion above, a literature analysis and discussion about the oxidation of propane to oxygenates was added in the revised manuscript, which may stress the novelty of this reaction with Pt SACs and make the Catalytic Evaluation section more comprehensive.

Page 10-11, line 243-251 “**The partial oxidation of propane to valuable liquid oxygenates represents a novel strategy to utilize this class of light alkane³⁴. Among the several current strategies, such as electrocatalysis³⁵, photocatalysis³⁶⁻³⁷, thermal-derived homogeneous³⁸, and thermal-derived heterogeneous catalysis, the exploitation of heterogeneous catalyst shows the greatest application potential³⁹⁻⁴¹. However, the consumption of costly oxidants poses an obstacle to it. To address this issue, a catalytic process that can transfer propane to oxygenates with low-cost oxidants is urgently needed. Inspired by the molecular oxygen activation capacity of isolated Pt sites², Pt SACs/PCN was evaluated in the oxidation of propane with oxygen in this work.**”

Moreover, the reaction rates of our catalysts were cast in terms of turnover frequencies (TOF) and compared with other reported catalysts, the results were listed in Table S4 and discussed in the revised manuscript.

Page 12, line 268-281, “As shown in Figure 5d, the liquid product is confirmed as 37.1 mmol/g_{cat} at 3h, which increases with the reaction time, and reaches 71.9 and 107.6 mmol/g_{cat} at 6 and 9 h, which surpasses the low-loading Pt SACs/PCN, Pt nanoparticles (Pt NPs/PCN) and commercial Pt/C catalyst (Figure 5e). To reveal the intrinsic activity of Pt SACs/PCN, the turnover frequency (TOF) and mass-specific activity are confirmed as $1.6 \times 10^{-3} \text{ mol}_{\text{pro}} \cdot \text{mol}_{\text{Pt}}^{-1} \cdot \text{s}^{-1}$ and 12.0 mmol/g_{cat}/h (Figure 5f-g and Table S4), superior to the reported propane activation performance with oxygen. Interestingly, among the catalysts that worked with oxygen, only Pt SACs/PCN the pathway toward oxygenates (Figure 5g). This may be the first investigation on the heterogeneous catalytic oxidation of propane to oxygenates with oxygen, which provides a novel stage to utilize propane for harvesting valuable liquid productions.”

Figure 5. (d) Catalytic performance of Pt SACs/PCN over reaction time. (e) Catalytic performance of various catalyst at 6h. (f) TOF value of the catalyst used in this work. (g) Comparison on the propane oxidation performance with previous reports. (h) Stability test. The maximum measurement error for (d) and (e) is $\pm 3.8\%$.

Table S4. Comparison on the catalytic performance of propane oxidation with molecule

oxygen and other oxidants.

	Catalyst	Oxidant	Production	Temp. (°C)	Mass activity (mmol/g _{cat} /h)	TOF (mol _{pro} /mol _M ⁻¹ s ⁻¹)	Ref.
1	41.8 % Pt SAC/PCN	O ₂	oxygenates	175	12.0	1.6 x 10 ⁻³	This work
2	17 % Pt SAC/PCN	O ₂	oxygenates	175	6.7	2.1 x 10 ⁻³	This work
3	34.1 % Pt SAC/NC	O ₂	oxygenates	175	6.1	0.96 x 10 ⁻³	This work
4	Cu powder dispersed in 1.0 M HClO ₄	O ₂	propylene	25	1.8	3.3 x 10 ⁻⁵	Nat. Catal. , 2023, 6, 666-675
5	TiO ₂	O ₂	CO ₂	UV	1.1	2.4 x 10 ⁻⁵	J. Catal. , 2015, 324, 119-126
6	LaCo _{0.1} Mn _{0.9} O ₃	O ₂	CO ₂	260	1.0	6.3 x 10 ⁻⁵	J. Phys. Chem. C , 2020, 124, 14646-14657
7	Immobilised iron complex	H ₂ O ₂	oxygenates	50	-	2.8 x 10 ⁻³	Catal. Sci. Technol. , 2023, 13, 4839-4846
8	Tricopper cluster complex	H ₂ O ₂	oxygenates	25	11.3	7.1 x 10 ⁻²	ACS Sustainable Chem. Eng. , 2018, 6, 5431-5440
9	CoCl _{1.6} Pc-Na-X (0.27)	TBHP and O ₂	oxygenates	25	-	1.7 x 10 ⁻²	Catal. Today , 1999, 49, 171-175
10	CuCl _{1.6} Pc-Na-Y (0.11)	TBHP and O ₂	oxygenates	25	-	1.3 x 10 ⁻²	
11	Ti(TFA) ₃ homogeneous	Ti(TFA) ₃	oxygenates	180	1.6	8.1 x 10 ⁻⁵	Science , 2014, 343, 1232-1237

4. Why was the reactivity of NC series of catalysts not evaluated? Showing a similar trend of activity would further boost the impact.

Answer: Thanks to the comment. The reactivity of Pt SACs/NC had been evaluated according to the suggestion of the reviewer, which follow the similar trend with that of Pt SACs/PCN. The results were showed in Figure 5e and discussed in the revised manuscript.

Page 12, line 281-285, “To clarify the effect of the substrates, Pt SACs on NC were also evaluated in the propane oxidation (Figure 5e). Following a similar trend with Pt SACs/PCN, Pt SACs/NC with higher Pt loading show better activity than those with lower Pt loading and Pt particles, and the productions are dominated by oxygenates.”

Figure 5. (e) Catalytic performance of various catalyst at 6h.

5. Given that the results of Figure 5e are central to the science, it is essential that these results are replicable. Has this been done? If not, it should be, and error bars should be added to the figure or at least commented on in the text.

Answer: Thanks to the comments. To reinforce the reliability of Figure 5d and e, the replicability of these results had been explored by repeating each catalytic process for at least three times under the same condition. The activity of each single catalyst

showed slight fluctuation, confirming the repeatability of this reaction process. Moreover, error bars had been added into Figure 5d-e.

Figure 5. (d) Catalytic performance of Pt SACs/PCN over reaction time. (e) Catalytic performance of various catalyst at 6h.

In sum, getting beyond the poor English, this paper was a thrill to read. The work appears to be a wonderful advance in synthesis science, combined with a new class of nitrogendoped carbons to allow atom isolation at previously unachievable areal loadings, and which show high activity for pertinent catalytic reactions. It should not be difficult to tighten up the loose ends in the manner suggested.

Reviewer #2 (Remarks to the Author):

The authors designed a generalized negative pressure annealing method for the preparation of low-cost, high-density, ultra-high-loaded single-atom catalysts (SACs) on carbon nitride substrates. Negative pressure and heat treatment lead to rapid dispersion rather than aggregation of metal atoms, which are subsequently captured by nitrogen sites, and ultrahigh metal loadings ranging from 27.3% to 44.8% can be achieved for multiple SACs on polymerized carbon nitride. In addition, high-entropy SACs with high metal content containing multiple metal single atoms have been obtained on carbon nitride, demonstrating the general applicability of the negative pressure annealing method. I would recommend this manuscript to be accepted with some revisions as mentioned in the following.

1. Please describe in detail the process of synthesizing Pt UHL-SACs by pyrolysis.

Answer: Thanks to the comment. The process of synthesizing Pt UHL-SACs by pyrolysis had been described detailly at the Methods section in main text of the revised manuscript.

“Methods

Synthesis of PCN. The mixture of melamine and dicyandiamide (molar ratio=7:3) was pyrolyzed in a tube furnace at 600 °C for 1.5 h (Ar flow, 100 sccm). The obtained powder (3 g) was treated in 65wt% HNO₃ (50 ml) at 80 °C for 6 h, followed by an ultrasonic treatment for 1h. The suspension was centrifuged and washed with deionized water to gain yellow powder, which was labelled as polymeric carbon nitride (PCN).

Synthesis of NC. Guanine was pyrolyzed in a tube furnace at 600 °C for 1.5 h (Ar flow, 100 sccm). The obtained black powder was labelled as N-doped carbon (NC).

Synthesis of Pt SACs/PCN, Pt NPs/PCN, and Pt SACs/NC. Pt SACs/PCN was prepared via the impregnation-vacuum pyrolysis method. 20 ml chloroplatinic acid solution (0.075 mol/L) was added in 1 g PCN. The obtained suspension was ultrasonic treated for 15 min, aged at room temperature for 2 h, and dried at 80 °C for 8 h. The obtained powder was placed in a tube furnace. The tube furnace was firstly purged by 100 sccm Ar flow for 15 min. Then, the outlet was connected to an operating mechanical pump (limiting pressure 6×10^{-2} Pa), and the inlet was closed. After 1 h of pyrolysis under vacuum condition at 400 °C, the black powder Pt SACs/PCN was obtained. Pt/NPs/PCN was prepared via the the same procedure, except for the pyrolysis process, which was carried out in Ar flow (atmospheric pressure). Pt SACs/NC was prepared by the impregnation-vacuum pyrolysis method, using 0.035 mol/L chloroplatinic acid solution and NC substrate.

Synthesis of M SACs/PCN and M SACs/NC. M SACs/PCN (M=V, Cr, Mn, Fe, Co, Ni, Cu, Zn, Nb, Mo, Ir and Au) was synthesized via the same method with Pt SACs/PCN, using vanadyl sulfate, chromic nitrate, manganous nitrate, ferric nitrate, cobaltous nitrate, nickel nitrate, cupric nitrate, zinc nitrate, niobium oxalate, molybdenum pentachloride, chloro-iridic acid, and tetrachloro-auric acid as the metal precursor, respectively. The concentration of metal precursor solution was 0.17 mol/L for V, Cr, Mn, 0.2 mol/L for Fe, Co, Ni, Cu and Zn, 0.12 mol/L for Nb and Mo, 0.075 mol/L for Ir, and 0.045 mol/L for Au. The vacuum pyrolysis temperature for V, Cr, Mn, Fe, Co, Ni, Cu and Zn SACs/PCN was 500 °C. M SACs/NC (M= Fe, Co, Ni, Cu) was

prepared according to the method of M SACs/PCN. The concentration of metal precursor solution was 0.07 – 0.1 mol/L.

Synthesis of HESACs. HESACs was prepared according to the method of M SACs/NC. The metal precursor solution was a mixture of ferric nitrate, cobaltous nitrate, nickel nitrate, cupric nitrate, and chloroplatinic acid. Their concentration in the solution was 0.015, 0.015, 0.01, 0.03, and 0.02 mol/L, respectively.

More details about the characterization and catalytic evaluation are presented in the supporting information files. ”

2. For discussing the adaptability of the negative pressure annealing method on different substrates, the authors have selected only five metals for single-atom doping. Whether the versatility of this method can be proved, please explain it.

Answer: Thanks to the comment. In this work, the adaptability of synthetic process should stem from the unique evolution pathway of metal species under negative pressure annealing condition. That is, the negative pressure condition accelerates the generation of isolated metal species with coordination ability, and suppresses the metal aggregation via promoting the metal-N coordination. In the negative pressure evolution, the pivotal role of the carrier is offering active sites to coordinate with isolated metal, such as N sites. This evolution pathway had been proved on the formation of Pt SACs on polymeric carbon nitride (PCN) carrier (Figure 2), and was furtherly used to prepare other 12 metal SACs on PCN (Figure 3). To evaluate the adaptability of the negative pressure evolution on substrates other than PCN, N-doped carbon (NC) was selected. Although the NC has different chemical natures with PCN, but it can also provide active N sites to form metal-N coordination for preparing multiple SACs (*Nat. Nanotech.*, 2022, **17**, 174-181 and *Angew. Chem. Int. Ed.*, 2021, **60**, 21751-21755). For NC support, the typical noble (Pt) and non-noble (Fe, Co, Ni, Cu) single atom catalysts were constructed through negative pressure annealing, considering their analogous property for others noble and non-noble metal species and widespread application for various catalysis. Although limited number of metals were tested, one can speculate that the formation of SACs on NC follow the same evolution pathway with that on PCN. In other words, more metal SACs can be prepared on NC, if the selected metal species can form metal-N coordination. Therefore, the versatility of this negative pressure annealing method on different N-containing substrates is foreseeable. Besides the reasonable conjecture

above, the high costs of spherical aberration correction transmission electron microscope and scarcity of XAS cause an obstacle for the comprehensive measurement of all metal SACs on NC.

To tighten up the loose ends about the versatility of the negative pressure annealing method, an explanation had been made in the revised manuscript:

Page 10, line 228-232, “Although a limited number of metals were tested on NC, we speculate that the formation of SACs on NC follows the same evolution pathway as that on PCN, demonstrating the versatility of this synthetic method on different N-containing carbon substrates for preparing SACs with multiple metals.”

3. The catalytic evaluation of Pt SACs/PCNs in Chapter 2.3 is not comprehensive enough, and more discussion is needed to demonstrate the excellent performance of SACs.

Answer: Thanks to the professional comments. To make the catalytic evaluation of Pt SACs/PCNs more comprehensive, the Chapter 2.3 had been reconstructed in the revised manuscript, including:

(1) Literature discussion about the novelty of this reaction with Pt SACs was added.

Page 10-11, line 243-251 “The partial oxidation of propane to valuable liquid oxygenates represents a novel strategy to utilize this class of light alkane³⁴. Among the several current strategies, such as electrocatalysis³⁵, photocatalysis³⁶⁻³⁷, thermal-derived homogeneous³⁸, and thermal-derived heterogeneous catalysis, the exploitation of heterogeneous catalyst shows the greatest application potential³⁹⁻⁴¹. However, the consumption of costly oxidants poses an obstacle to it. To address this issue, a catalytic process that can transfer propane to oxygenates with low-cost oxidants is urgently needed. Inspired by the molecular oxygen activation capacity of isolated Pt sites², Pt SACs/PCN was evaluated in the oxidation of propane with oxygen in this work.”

(2) The catalytic performance of Pt SACs on NC substrate was evaluated and listed in Figure 5e, and the TOF and mass activities of various catalysts used in this work were presented in Figure 5f-g and Table S4, and compared with other reported catalysts. The above results are discussed in the revised manuscript.

Page 12, line 268-287, “As shown in Figure 5d, the liquid product is confirmed as 37.1 mmol/g_{cat} at 3h, which increases with the reaction time, and reaches 71.9 and 107.6 mmol/g_{cat} at 6 and 9 h, which surpasses the low-loading Pt SACs/PCN, Pt nanoparticles

(Pt NPs/PCN) and commercial Pt/C catalyst (Figure 5e). To reveal the intrinsic activity of Pt SACs/PCN, the turnover frequency (TOF) and mass-specific activity are confirmed as $1.6 \times 10^{-3} \text{ mol}_{\text{pro}} \cdot \text{mol}_{\text{Pt}}^{-1} \cdot \text{s}^{-1}$ and $12.0 \text{ mmol/g}_{\text{cat}}/\text{h}$ (Figure 5f-g and Table S4), superior to the reported propane activation performance with oxygen. Interestingly, among the catalysts that worked with oxygen, only Pt SACs/PCN the pathway toward oxygenates (Figure 5g). This may be the first investigation on the heterogeneous catalytic oxidation of propane to oxygenates with oxygen, which provides a novel stagey to utilize propane for harvesting valuable liquid productions. Moreover, the catalytic performance of Pt SACs/PCN shows insignificant decay after being reused five times (Figure 5h), and the used catalyst maintains the dense isolated Pt sites (Figure S46), confirming its stability. To clarify the effect of the substrates, Pt SACs on NC were also evaluated in the propane oxidation (Figure 5e). Following a similar trend with Pt SACs/PCN, Pt SACs/NC with higher Pt loading show better activity than those with lower Pt loading and Pt particles, and the productions are dominated by oxygenates. These catalyst evaluations demonstrate the potential application of high-loading Pt SACs in activating light alkanes.”

Figure 5. (d) Catalytic performance of Pt SACs/PCN over reaction time. (e) Catalytic performance of various catalyst at 6h. (f) TOF value of the catalyst used in this work. (g) Comparison on the propane oxidation performance with previous reports. (h) Stability test. The maximum measurement error for (d) and (e) is $\pm 3.8\%$.

Table S4. Comparison on the catalytic performance of propane oxidation with molecule

oxygen and other oxidants.

	Catalyst	Oxidant	Production	Temp. (°C)	Mass activity (mmol/g _{cat} /h)	TOF (mol _{pro} /mol _M ¹ ·s ⁻¹)	Ref.
1	41.8 % Pt SAC/PCN	O ₂	oxygenates	175	12.0	1.6 x 10 ⁻³	This work
2	17 % Pt SAC/PCN	O ₂	oxygenates	175	6.7	2.1 x 10 ⁻³	This work
3	34.1 % Pt SAC/NC	O ₂	oxygenates	175	6.1	0.96 x 10 ⁻³	This work
4	Cu powder dispersed in 1.0 M HClO ₄	O ₂	propylene	25	1.8	3.3 x 10 ⁻⁵	Nat. Catal. , 2023, 6, 666-675
5	TiO ₂	O ₂	CO ₂	UV	1.1	2.4 x 10 ⁻⁵	J. Catal. , 2015, 324, 119-126
6	LaCo _{0.1} Mn _{0.9} O ₃	O ₂	CO ₂	260	1.0	6.3 x 10 ⁻⁵	J. Phys. Chem. C , 2020, 124, 14646-14657
7	Immobilised iron complex	H ₂ O ₂	oxygenates	50	-	2.8 x 10 ⁻³	Catal. Sci. Technol. , 2023, 13, 4839-4846
8	Tricopper cluster complex	H ₂ O ₂	oxygenates	25	11.3	7.1 x 10 ⁻²	ACS Sustainable Chem. Eng. , 2018, 6, 5431-5440
9	CoCl _{1.6} Pc-Na-X (0.27)	TBHP	oxygenates	25	-	1.7 x 10 ⁻²	Catal. Today , 1999, 49, 171-175
10	CuCl _{1.6} Pc-Na-Y (0.11)	and O ₂	oxygenates	25	-	1.3 x 10 ⁻²	
11	Ti(TFA) ₃ homogeneous	Ti(TFA) ₃	oxygenates	180	1.6	8.1 x 10 ⁻⁵	Science , 2014, 343, 1232-1237

4. Some classic or recent references about SAC s also need to be added, such as *Nature*, 2023, (622):754-760, *Nano Energy*, 2023, (111): 108404, *Sci. China Mater.*, 2023, 66(3): 1080-1089, and *J Mater. Chem. A*, 2022, (10): 6231-6241.

Answer: Thanks to the comments. These works had been cited and discussed in the revised manuscript.

Page 3, line 54-57, “Single atom catalysts (SACs), integrating atomically dispersed metal center with tunable coordination structure, exhibit remarkable activity and unique selectivity for electrocatalysis, photocatalysis, thermos-catalysis²⁻⁷. Moreover, the maximal atom utilization efficiency of this class of catalysts greatly improves the atom economy, especially for noble-metal-based catalysts, therefore beneficial for sustainable chemistry⁸⁻¹².”

6. Q. Yang, Y. Jiang, H. Zhuo, E. M. Mitchell, Q. Yu, Recent Progress of Metal Single-Atom Catalysts for Energy Applications, *Nano Energy*, 2023, 11, 108404.

7. P. Rong, Y.-F. Jiang, Q. Wang, M. Gu, X.-L. Jiang, Q. Yu, Photocatalytic Degradation of Methylene Blue (MB) with Cu₁-ZnO Single Atom Catalysts on Graphene-Coated Flexible Substrates, *J. Mater. Chem. A*, 2022, 10, 6231-6241.

11. X. Hai, Y. Zheng, Q. Yu, N. Guo, S. Xi, X. Zhao, S. Mitchell, X. Luo, V. Tulus, M. Wang, X. Sheng, L. Ren, X. Long, J. Li, P. He, H. Lin, Y. Cui, X. Peng, J. Shi, J. Wu, C. Zhang, R. Zou, G. Guillén-Gosálbez, J. Pérez-Ramírez, M. J. Koh, Y. Zhu, J. Li, J. Lu, Geminal-Atom Catalysis for Cross-Coupling, *Nature*, 2023, 622, 754-760.

12. Q. Yu, Theoretical Studies of non-Noble Metal Single-Atom Catalyst Ni₁/MoS₂: Electronic Structure and Electrocatalytic CO₂ Reduction, *Sci. China Matter.*, 2023, 66, 1079-1088.

5. Details need to be adjusted, e.g., N-doped Carbon should be annotated with the full

name on the first occurrence, and the same phrase should be replaced by an acronym on subsequent occurrences. The quality of the pictures is not high enough, and the resolution of the graphs should be increased (such as Fig. 1 and Fig. 3). Too many pictures are put in Fig. 4, resulting in the pictures not being clear enough. It is recommended that Fig. 4 be readjusted.

Answer: Thanks to the nice comments. The issue of acronym had been revised. The resolution of Figure 1 and 3 had been improved. The Figure 4 had been readjusted to make it clearer. Moreover, other details including the language had also been revised.

Figure 1. Structure investigation of the Pt UHL-SACs (Pt SACs/PCN). (a) Metal contents of the most reported Pt SACs. (b) Schematic illustration for preparing UHL-SACs. (c) TEM images of 40.9 wt% Pt NP/PCN. (d) XRD patterns of 40.9 wt% Pt NP/PCN and 41.8 wt% Pt SACs/PCN. (e-n) Structure characterization of 41.8 wt% Pt SACs/PCN: (e) TEM image, (f-h) Aberration-corrected HAADF-TEM image and the corresponding intensity profiles in the (g) yellow square and (h) red square, (i-j) EDS element mapping, (k) XPS results of 41.8 wt% Pt SACs/PCN. (l) Pt L-edge XANES

spectra, (m) Pt *L*-edge FT EXAFS spectra and (n) the corresponding wavelet transformation results.

Figure 3. The universal preparation of UHL-SACs on PCN (M SACs/PCN). (a) The metal elements used for fabricating M SACs/PCN. **(b)** The metal content in the as prepared catalysts. **(c)** Aberration-corrected HAADF-STEM and **(d)** FT-EXAFS spectra of various M SACs/PCN.

Figure 4. Fabricating UHL-SACs on N-doped Carbon (NC) substrate. The aberration-corrected HAADF-STEM images and FT-EXAFS spectra of NC supported UHL-SACs with (a) Pt, (b) Fe, (c) Co, (d) Ni, and (e) Cu. (f) TEM and (g) Aberration-corrected HAADF-STEM images of the UHL HESACs. (h) The metal content in the HESACs. (i) FT-EXAFS spectra for the metals in HESACs.

REVIEWERS' COMMENTS

Reviewer #1 (Remarks to the Author):

I congratulate the authors on a very interesting and provocative paper. All the changes I had suggested were thoroughly made. The extremely high areal loading of several of the metals (over 20 atoms/nm²!) was explained by percolation into the bulk. This deserves more attention in future work, in my opinion.

In the revised sections there are yet more grammatical errors, but those can be forgiven in view of the overall work. I recommend publication as is.

I can imagine the impact of the work will be quite substantial. Thanks for opening up this method!

Reviewer #2 (Remarks to the Author):

After check the revision, the author has corrected the referee's problems, thus, I recommend to accept the paper for publish.